# Adsorption and Release of Rose Bengal on Layer-by-Layer Films of Poly(Vinyl Alcohol) and Poly(Amidoamine) Dendrimers Bearing 4-Carboxyphenylboronic Acid

**DOI:** 10.3390/polym12081854

**Published:** 2020-08-18

**Authors:** Kentaro Yoshida, Akane Yamaguchi, Hiroki Midorikawa, Toshio Kamijo, Tetsuya Ono, Takenori Dairaku, Takaya Sato, Tsutomu Fujimura, Yoshitomo Kashiwagi, Katsuhiko Sato

**Affiliations:** 1School of Pharmaceutical Sciences, Ohu University, 31-1 Misumido, Tomita-machi, Koriyama, Fukushima 963-8611, Japan; 715078@ohu-u.jp (A.Y.); 715070@ohu-u.jp (H.M.); t-ono@pha.ohu-u.ac.jp (T.O.); t-dairaku@pha.ohu-u.ac.jp (T.D.); y-kashiwagi@pha.ohu-u.ac.jp (Y.K.); 2Department of Creative Engineering, National Institute of Technology, Tsuruoka College, 104 Sawada, Inooka, Tsuruoka, Yamagata 997-8511, Japan; kamijo@tsuruoka-nct.ac.jp (T.K.); takayasa@tsuruoka-nct.ac.jp (T.S.); satok@tohoku-mpu.ac.jp (K.S.); 3Faculty of Pharmaceutical Science, Tohoku Medical and Pharmaceutical University, 4-4-1 Komatsushima, Aoba, Sendai, Miyagi 981-8558, Japan; tfujitsu@tohoku-mpu.ac.jp

**Keywords:** layer-by-layer films, dendrimer, boronic acid, release system, thin film

## Abstract

Phenylboronic acid-bearing polyamidoamine dendrimer (PBA-PAMAM)/poly(vinyl alcohol) (PVA) multilayer films were prepared through the layer-by-layer (LbL) deposition of PBA-PAMAM solution and PVA solution. PBA-PAMAM/PVA films were constructed successfully through the formation of boronate ester bonds between the boronic acid moiety in PBA and 1,3-diol units in PVA. When the (PBA-PAMAM/PVA)_5_ films were immersed in rose bengal (RB) solution, RB was adsorbed onto the LbL films. The amount of RB adsorbed was higher in the LbL films immersed in acidic solution than in basic solution. The release of RB from the LbL films was also promoted in the basic solution, while it was suppressed in the acidic solution. The boronic acid ester is oxidized to phenol by hydrogen peroxide (H_2_O_2_) and the carbon-boron bond is cleaved, so that the (PBA-PAMAM/PVA)_5_ films can be decomposed by immersion in H_2_O_2_ solution. Therefore, when RB-adsorbed (PBA-PAMAM/PVA)_5_ films were immersed in H_2_O_2_ solution, the release of RB was moderately promoted when the solution was weakly acidic.

## 1. Introduction

Layer-by-layer (LbL) films have been developed for separation membranes [1,2], sensors [3,4,5], and capsules as controlled release systems [6,7,8,9]. Such LbL films are prepared using various polymers for the development of thin film devices. Examples of the polymer materials used include synthetic polymers [10,11], polysaccharides [12,13,14], proteins [15,16], and DNA [17,18,19]. Thin films with various functions can be developed using polymers that are suitable for the intended application. In this context, we have focused on the preparation of thin films using dendrimers. Poly(amidoamine) (PAMAM) dendrimers are highly branched spherical nanostructures and are expected to be applied in drug delivery systems [20,21].

LbL films can be prepared by the alternate deposition of different polymers on a solid surface using electrostatic interaction [22,23], hydrogen bonding [24,25], sugar-lectin bonding [26], and host-guest complexation [27]. In particular, many LbL films prepared using electrostatic interaction as a driving force have been reported. However, the adsorption of drugs in thin films can also be affected by macromolecules of the same charge if electrostatic interactions are used. We have focused on interactions that can form a thin film with a driving force other than electrostatic interaction. Among these interactions, phenylboronic acid (PBA) derivatives are known to bind to compounds bearing 1,2- and 1,3-diol moieties by the formation of boronate ester bonds in aqueous solutions (Figure 1) [28,29]. Suwa et al. recently reported that LbL films were prepared by combining PBA-bearing-PAMAM (PBA-PAMAM) and poly(vinyl alcohol) (PVA) [30]. PBA-PAMAM/PVA films were constructed successfully through the formation of boronate ester bonds between the boronic acid moiety in PBA and 1,3-diol units in PVA. The LbL films composed of PBA derivatives are decomposed in glucose and fructose solutions, depending on the glucose and fructose concentration, due to competitive binding of the sugars to PBA in the film [31]. The boronate ester is oxidized to phenol by hydrogen peroxide with cleavage of the carbon-boron bond [32,33] (Figure 2). PBA derivatives with sugar- and H_2_O_2_-responsiveness have utility in sensing blood glucose levels [34] and reactive oxygen species (ROS) in cells [35]. Therefore, if the LbL films using PBA-PAMAM can be decomposed in response to stimuli, they could be applied as a drug reservoir for controlled drug release systems. Here, we investigate the functionality of this thin film system using rose bengal (RB) dye as a model drug (Figure 3). We report here on the release of dye due to the change in pH, H_2_O_2_ concentration, and ionic strength by the adsorption of RB on PVA/PBA-PAMAM films.

## 2. Materials and Methods

### 2.1. Materials

PVA (typical MW 89,000–98,000) and PAMAM dendrimer (4th generation, ethylenediamine core, 10% methanol solution) were purchased from Sigma-Aldrich Co. (St. Louis, MO, USA). RB and 4-carboxyphenylboronic acid was obtained from Tokyo Kasei Co. (Tokyo, Japan). Then, 1-Ethyl-3-(3-dimethylaminopropyl)carbodiimide hydrochloride (EDC) and *N*-hydroxysuccinimide (NHS) were obtained from Nacalai Tesque Inc. (Kyoto, Japan).

PBA-PAMAM was synthesized by reaction of the dendrimer and 4-carboxyphenylboronic acid NHS ester in water, according to a reported procedure [36]. Thereafter, 4-PBA residues were attached to 39 mol% of the primary amino groups on the surface of PAMAM. All other reagents were of the highest grade and used without further purification. The structures of PBA-PAMAM, PVA, and RB are shown in Figure 4.

### 2.2. Preparation of LbL Films

LbL films were prepared on the quartz slide surfaces (50 by 9 by 1 mm) and circular glass slide surfaces (18 mm diameter) that had been cleaned in piranha solution (H_2_O_2_:H_2_SO_4_ = 1:3 *v/v*). The quartz slides were alternately immersed in a 0.1 mg/mL PBA-PAMAM solution and 0.1 mg/mL PVA solution in 10 mM *N*-Cyclohexyl-2-aminoethanesulfonic acid (CHES) buffer containing 150 mM NaCl (pH 9) for 15 min. The quartz slides and circular glass slides were rinsed with working buffer for 5 min to remove any weakly adsorbed PBA-PAMAM and PVA.

The deposition steps were repeated to build up (PBA-PAMAM/PVA)_5_ LbL films. UV-vis absorption spectra of the LbL films in the CHES buffer were recorded on a UV-vis spectrometer (V-560, Jasco, Tokyo, Japan, and UV3100PC, Shimadzu Co., Kyoto, Japan). The quartz resonators for quartz crystal microbalance (QCM; eQCM 10M, Garmry, Warminster, UK) analysis of the LbL films were prepared in the same manner. For dry atomic force microscopy (AFM; AFM5200S, Hitachi High-Technologies Co., Tokyo, Japan) observations, the circular glass slides used to prepare each of the (PBA-PAMAM/PVA)_5_ LbL films were rinsed with milli-Q water and dried for 24 h in a desiccator. AFM images were acquired in the AC mode using an Arrow-NCR probe (Toyo Corporation, Tokyo, Japan) at room temperature in air.

### 2.3. Adsorption and Release of RB on LbL Films

The quartz slides coated with LbL films were immersed in 0.1 mg/mL RB solution for 15 h and were then rinsed in the working buffer for 5 min to remove any weakly adsorbed RB. To evaluate the amount of adsorbed RB, the film was decomposed with 100 mM H_2_O_2_ solution (pH 9.0) for 1 h because the diol bond of PBA is cleaved by H_2_O_2_ [32,33], which induces decomposition [36]. The amount of RB adsorbed on the film was calculated by measuring the absorbance of the solution in which the LbL films was decomposed.

RB release from LbL films was evaluated using UV-vis absorption spectroscopy. The quartz slides coated with LbL films containing RB were exposed to various pH solutions for 5, 10, 15, 30, 45, 60, 90, 120, 150, and 180 min. UV-vis absorption of the various pH solutions with immersed LbL films was measured at a particular time and the LbL was then subsequently immersed in the working solution for the next exposure step. The total amount of RB adsorbed on the films was measured by immersion of the membrane in hydrogen peroxide solution (pH 9) for 12 h. The amount of RB was determined from the absorption spectrum of RB at the pH of the working buffers.

The buffers used were 10 mM acetate buffer (pH 4 and pH 5), 10 mM 2-(*N*-morpholino)ethanesulfonic acid (MES) buffer (pH 6), 10 mM 4-(2-hydroxyethyl)-1-piperazineethanesulfonic acid (HEPES) buffer (pH 7), and 10 mM CHES buffer (pH 8 and pH 9). All buffers contained 150 mM NaCl.

The release of RB due to changes in the H_2_O_2_ and NaCl concentrations was evaluated in a similar manner. The buffer used was a working buffer (pH 4, pH 7, and pH 9) containing 0, 1, 10, or 100 mM H_2_O_2_, and 10 mM HEPES (pH 7) containing 0, 10, 150, or 1000 mM NaCl.

## 3. Results and Discussion

### 3.1. Preparation of LbL Films

A quartz slide was alternately immersed in the PBA-PAMAM and the PVA solutions, and the absorbance of the quartz slide was measured each time (Figure 5). When the quartz plate was immersed in the PBA-PAMAM solution, absorption was observed at 250 nm. PBA has an absorption at 250 nm. The PBA-PAMAM was adsorbed on the quartz slide, and the PBA-PAMAM/PVA bilayer had an absorption maximum derived from PBA. The absorption at 250 nm increased with deposition of both PBA-PAMAM and PVA, which indicated that the (PBA-PAMAM/PVA)_5_ film was successfully formed on the surface of the quartz slide. PBA forms a boronate ester bond with diol compounds [28,29]. The diol bonding affinity toward PBA is strongly dependent on the pH because formation of the tetragonal boronate ester bonds occurs under basic conditions. The preparation of LbL films using a diol bond has been reported [30]. When preparing a membrane, a polymer using a diol bond is more likely to be deposited in basic solution, rather than acidic or neutral solution. It is noted that the change of absorbance at 250 nm is not linear with respect to the number of bilayers. Therefore, adsorption of PBA-PAMAM is higher for the outer layers than the inner layers. The loop or tail conformation of PVA on the surface may also contribute to the higher adsorption of PBA-PAMAM. This type of adsorption behavior has been observed in such LbL films as those composed of polypeptides and polysaccharides [37,38]. As a result, a larger number of bilayers results in a larger area where the polymer is adsorbed; therefore, the film grows non-linearly.

Figure 6 shows an AFM image and a depth profile of a dried (PBA-PAMAM/PVA)_5_ LbL film. The circular slides (18 mm diameter) used to prepare (PBA-PAMAM/PVA)_5_ films were rinsed with milli-Q water and dried for 24 h in a desiccator. The thickness of the LbL films was determined by scratching the film-coated glass slide using a cutter. The deep line in the AFM image is such a scratch, near which the film and the surface of the glass plate can be observed. AFM depth profile scans over the scratch indicated the thicknesses of the (PBA-PAMAM/PVA)_5_ film was approximately 46.9 ± 8.9 nm. The surface of thin films of one, three, and five bilayers was confirmed using AFM (Appendix A). Small depressions and aggregations were observed on the surface of the thin film of one-bilayer. This may be caused by polymer–substrate interactions due to the adsorption of the first polymer layer. In addition, as the number of bilayers increased, the deposition of large aggregates was observed on the surface of the LbL film, which also increased the arithmetic mean roughness of the film (in the case of one, three, and five bilayers, Sa was 2.38, 3.04, and 9.12 nm). This is related to the non-linear growth of the LbL film.

Figure 7A shows the change in the resonance frequency (ΔF) of the QCM when the quartz resonator was immersed in the PBA-PAMAM and PVA solutions. ΔF decreased with the deposition of PBA-PAMAM and PVA, which indicated that the (PBA-PAMAM/PVA)_5_ films were successfully formed on the surface of the quartz resonator. The change in the resonance frequency of the (PBA-PAMAM/PVA)_5_ films from the flow QCM data was −2414 ± 183 Hz (n = 3). Figure 7B shows ΔF for the (PBA-PAMAM/PVA)_5_ films prepared on the surface of the quartz resonator and immersed in different pH solutions. There was no significant change in the resonant frequency when the membrane was immersed in pH 9 and pH 8 buffers. On the other hand, when the membrane was immersed in a buffer solution with a pH of 7 or less, a decrease in the resonance frequency was observed. Suwa et al. reported that (PBA-PAMAM/PVA)_5_ films were maintained in neutral solution, whereas they degraded in weakly acidic solution [30]. However, there were differences in the compositions of the LbL films. While Suwa et al. [30] used PVA where the degree of polymerization was 500, PVA with a weight average molecular weight of 89,000–98,000 (polymerization degree > 2000) was used. A longer PVA chain was, thus, used in this study, which led to different results. Trigonal boronate ester bonds, which are predominant in acidic media, are less stable than boronate ester bonds with the tetragonal form. The interaction that forms the film may be explained based on an increase in the number of trigonal boronate ester bonds in the film. It should be noted that the (PBA-PAMAM/PVA)_5_ films were stable under weakly basic conditions, whereas weakly acidic solutions weaken the ester bonds that form the membrane. Shorter PVA chains are more likely to induce degradation of the LbL films. The number of ester bonds in the boronic acid may increase as the PVA chain becomes larger; therefore, if the boronic acid ester bonds necessary to form the LbL film remain, decomposition of the LbL film is not induced. However, the LbL films are less stable in acidic solutions, and the PVA chains in the film can be degraded. This degradation of the PVA chains may result in swelling of the LbL films. In addition, it is considered that the structural change of PAMAM with pH change also affects the swelling of the LbL film. Liu et al. reported that PAMAM was fully deprotonated at high pH (>10). The primary amines of PAMAM were protonated at neutral pH (approximately 7); primary and tertiary amines are protonated at low pH (<5) [39]. However, PAMAM undergoes pH-responsive conformational changes without swelling. On the other hand, these PAMAMs are solvated with water molecules and Cl^−^ counterions. LbL films prepared at pH 9, thus, have poor protonation of the primary amines, whereas the amines are protonated at lower pH 9 and the resonance frequency decreases due to solvation by water and Cl^−^.

### 3.2. Adsorption of RB on LbL Films

The adsorption of RB on the (PBA-PAMAM/PVA)_5_ films was evaluated using UV-vis absorption spectroscopy. Figure 8 shows the UV-vis spectra after immersion of the films coated on quartz plates in RB solutions of various pH and then rinsing with the same pH buffer as the RB solution. UV-vis spectra were measured using the same pH buffer as the RB solution. RB has absorptions at 526 nm and 558 nm (there is no change in the absorption spectrum of RB with pH in the visible region). The (PBA-PAMAM/PVA)_5_ films immersed in RB solution had spectra similar to RB. It has been reported that RB is included in dendrimers [40]. Therefore, RB could be adsorbed on the PAMAM of the LbL films. The RB derived absorbance increased in the LbL films immersed in acidic solution, in contrast to that immersed in the basic solution. The primary amines of PAMAM were protonated at neutral pH (approximately 7); primary and tertiary amines are generally protonated at low pH (<5) [39]. It is considered that RB with a negative charge is easily adsorbed by PAMAM with a positive charge. However, the LbL films immersed in pH 4 solution had very little absorbance derived from RB. The acid dissociation constant of RB is pKa = 4.7 [41]; therefore, the net negative charge is significantly reduced at pH 4 compared to that under basic conditions. Therefore, adsorption between PAMAM and RB becomes difficult.

Appendix A shows the change in ΔF of the QCM when the quartz resonator coated with (PBA-PAMAM/PVA)_5_ film was immersed in RB solutions with various pH. ΔF decreased with the adsorption of RB, and the amount of change in ΔF increased with a decrease in the pH; however, the amount of change in ΔF was smaller than that in the solution without RB. PAMAM is solvated by water molecules and Cl^−^ counterions; therefore, the adsorption of RB on the LbL film may have changed the solvation of PAMAM. It is thus possible that the viscoelastic properties of the LbL film coated-quartz resonator surface were changed significantly. For this reason, the amount of RB adsorbed was determined by decomposition of the thin film with adsorbed RB and the solution used at that time was measured using UV-vis absorption spectroscopy.

It has been reported that hydrogen peroxide (H_2_O_2_) decomposes LbL films by interaction with the boronic acid ester bonds [32,33,36]. When the LbL film decomposes, adsorbed RB is released into the solvent (Appendix A). The surface of the LbL film immersed in H_2_O_2_ solution became smooth (Sa values of the film before and after immersion in H_2_O_2_ solution were 9.12 and 1.67 nm, respectively), and most of the film was decomposed (Appendix A). The LbL films were immersed overnight in RB solutions (0.1 mg/mL) with various pH, and the RB-adsorbed (PBA-PAMAM/PVA)_5_ films were then immersed in 100 mM H_2_O_2_ solution (pH 9). These solutions were then used to measure the total amount of RB that was adsorbed on the LbL films using UV-vis spectroscopy (Figure 9). The amount of RB adsorbed was higher in the LbL films immersed in the acidic solution than in the basic solution, and the amount of adsorbed RB on the LbL films immersed in the pH 4 solution was significantly low. The results in Figure 9 show similar trends to those in Figure 8. Therefore, it is considered that the amount of RB adsorbed differs between the structure of the LbL films and the charge of RB. The QCM results showed that the resonance frequency decreased when the (PBA-PAMAM/PVA)_5_ films were taken from basic solution and immersed into acidic solution (Figure 7B). This is because the diol bond in the LbL film is weakened, and swelling of the LbL films occurs by the uptake of water from the solvent. The swollen LbL film facilitates the adsorption of RB onto PAMAM. In addition, the charge of the dendrimer primary amines and boronic acid also affects RB adsorption. In particular, PAMAM with a positive charge adsorbs RB with a negative charge by electrostatic interaction. The amount of protonated primary and tertiary amines of PAMAM increases at lower pH than at higher pH. PAMAM thus undergoes protonation and conformational changes due to pH changes, which results in a difference in the amount of RB adsorbed. It is also necessary to consider the charge of RB. The acid dissociation constant of RB is pKa = 4.7 [41]; therefore, the net negative charge is significantly reduced at pH 4 compared to that under basic conditions. Under conditions of pH 4, the electrostatic interaction between PAMAM and RB is thus weakened, and the amount of RB adsorbed on the LbL films is significantly reduced.

### 3.3. Release of RB on LbL Films

RB-adsorbed (PBA-PAMAM/PVA)_5_ films were immersed in buffer solutions of various pH, and the release of RB from the LbL films was estimated (Figure 10). The slow release of RB continued in the same buffer solution (pH 7) as the RB solution for LbL film immersion. The adsorption of RB on the LbL films at pH 7 is due to electrostatic interaction; however, RB was released from the LbL films because it was not strongly adsorbed. Under basic conditions (pH 9), the stability of the LbL films is improved and the net positive charge from PAMAM may be weakened. Therefore, it is considered that the release of RB was promoted. In contrast, RB release was suppressed in acidic solution (pH 4 and pH 5). It is likely that under acidic conditions, the swelling of the LbL films increases the thickness of the LbL film. The primary and tertiary amines of PAMAM are protonated at low pH (<5). As a result, RB may be easily adsorbed to PAMAM. It is possible that these increased adsorption sites of RB could suppress the release of RB. In addition, PAMAM shows pH-induced conformational changes such as a dense core at high pH and a dense shell at low pH [39]. The pH-induced conformational change of PAMAM may, thus, be dependent on the adsorption and release of RB.

RB-adsorbed (PBA-PAMAM/PVA)_5_ films were immersed in buffer solutions with various H_2_O_2_ concentrations, and the release of RB from the LbL films was estimated. Under neutral conditions (pH 7), RB release was sustained slowly, and the RB release rate increased slightly in the presence of H_2_O_2_ for 15–90 min (Figure 11A). The increase in RB release is due to the decomposition of the LbL films by H_2_O_2_. However, after 120 min, there was no difference in the RB release rate with the different H_2_O_2_ concentrations. The RB release was also more sensitive to pH than to the presence of H_2_O_2_. To avoid the effects of RB release under neutral conditions, RB release in H_2_O_2_ solution was measured under acidic conditions (pH 4) (Figure 11B). The release of RB was suppressed in acidic solution, whereas, it was promoted in the presence of H_2_O_2_. Even under acidic conditions, there was no difference in the RB release rate due to the difference in H_2_O_2_ concentration. The decomposition of the RB-adsorbed (PBA-PAMAM/PVA)_5_ films was observed using QCM. Appendix A shows the change in ΔF when the (PBA-PAMAM/PVA)_5_ film and RB adsorbed-(PBA-PAMAM/PVA)_5_ film were immersed in various concentrations of H_2_O_2_. When the (PBA-PAMAM/PVA)_5_ film was immersed in 10 mM H_2_O_2_, a change in ΔF due to LbL film decomposition was observed. On the other hand, the change ΔF when the RB adsorbed-(PBA-PAMAM/PVA)_5_ film was immersed in H_2_O_2_ solution was low. The adsorption of RB may suppress the decomposition of the LbL by H_2_O_2_. It is possible that RB is deposited on the film due to the reduced solubility of RB under acidic conditions and the adsorption between RB and PAMAM. Furthermore, ΔF is more moderately increased in acid compared to neutral solution. The rate of decomposition of the LbL films by H_2_O_2_ may be slower in acidic than neutral solutions. The release rate of RB is slow because of the slow rate of decomposition of the LbL films by H_2_O_2_. However, the thin film immersed in H_2_O_2_ under acidic conditions shows a gradual increase in ΔF. Therefore, it may be that the dissolution and diffusion of RB was gradually promoted by the decomposition of the LbL film in H_2_O_2_.

The adsorption onto and release of RB from the (PBA-PAMAM/PVA)_5_ films may be modified by electrostatic interaction between PAMAM and RB. For this reason, an increase of the ionic strength of the associated solutions was assessed as a means to suppress the adsorption of RB and promote the release of RB. The LbL films were immersed in RB solutions (0.1 mg/mL) with various NaCl concentrations (0, 10, 150, and 1000 mM).

The amount of RB adsorbed was higher in the LbL films immersed in acidic solution than in basic solution; however, there was no significant dependence of the amount of RB adsorption the NaCl concentration (Appendix A). RB-adsorbed (PBA-PAMAM/PVA)_5_ films were immersed in buffer solutions with various NaCl concentrations, and the release of RB from the LbL films was estimated (Appendix A). Similar to the amount of RB adsorbed, there was no dependence of the release of RB from the LbL film on the NaCl concentration.

## 4. Conclusions

PBA-PAMAM/PVA LbL films were prepared by alternate immersion of a substrate in PBA-PAMAM solution and PVA solution. The resonance frequency was decreased in acidic solution compared to that in basic solution when the (PBA-PAMAM/PVA)_5_ films prepared on the surface of a quartz resonator were immersed. Trigonal boronate ester bonds, which are predominant in acidic media, are less stable than the tetragonal form. Therefore, the diol bond, which is the driving force necessary to form the film, is weakened. When the affinity between PVA and PBA-PAMAM weakens, the PVA chains inside the LbL films degrade. The degraded PVA chains may have caused the LbL films to become swollen. When (PBA-PAMAM/PVA)_5_ films were immersed in RB solution, RB was adsorbed on the LbL films. The amount of RB adsorbed on the thin films was higher in weakly acidic solution than in basic solution. On the other hand, the amount of adsorbed RB was significantly low at pH 4. RB-adsorbed (PBA-PAMAM/PVA)_5_ films were immersed in buffer solutions with various pH; the RB release was accelerated in basic solution and suppressed in acidic solution. PAMAM undergoes pH-induced changes in protonation and conformation, which are related to the adsorption and release of RB. Therefore, PAMAM is useful as a drug reservoir and can be used to control the entrapment and release of guest molecules (such as drugs) by pH induction. LbL films can be applied as coating for nanoparticles and microcapsule films [6,7,8,9]. Application as a drug reservoir can be expected by modifying the shape of the PBA-PAMAM/PVA film. For example, the inhibition of drug release under acidic conditions and the promotion of drug release in neutral solutions has potential applications in enteric coatings. RB release from the LbL films was slightly promoted when RB-adsorbed (PBA-PAMAM/PVA)_5_ films were immersed in H_2_O_2_ solution. H_2_O_2_ can be produced using glucose oxidase and lactate oxidase. It is possible that the LbL film could, thus, be decomposed in response to glucose and lactic acid. Furthermore, it has been reported that the pH around tumor tissue is weakly acidic and ROS are generated [35,42,43]. If this film can be optimized, then it may be possible to change the sustained-release rate of the drug by causing the film to decompose, depending on the amount of ROS around tumor tissue.

## Figures and Tables

**Figure 1 polymers-12-01854-f001:**
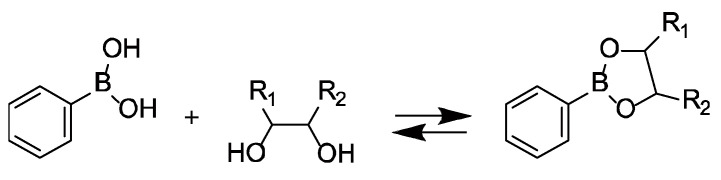
Bonding equilibrium between phenylboronic acid (PBA) and diol.

**Figure 2 polymers-12-01854-f002:**
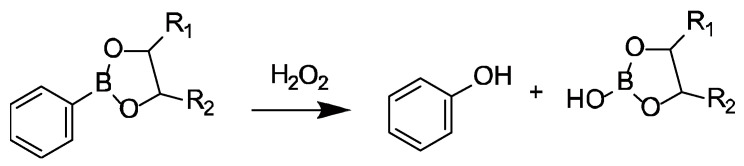
Boronate ester bond cleavage induced by H_2_O_2_.

**Figure 3 polymers-12-01854-f003:**
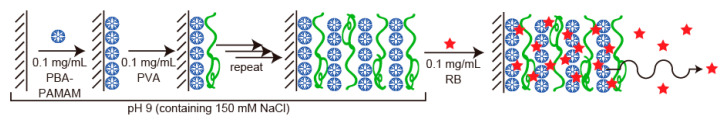
Schematic illustration of PBA-bearing- poly(amidoamine) (PBA-PAMAM)/poly(vinyl alcohol) (PVA) layer-by-layer (LbL) film preparation and rose bengal (RB) adsorption.

**Figure 4 polymers-12-01854-f004:**
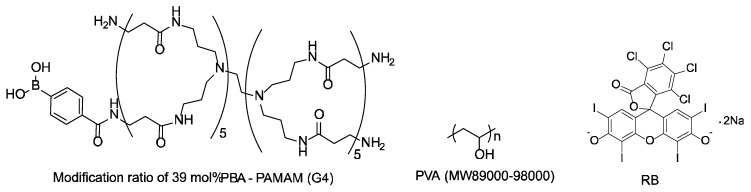
Chemical structures of PBA-PAMAM, PVA, and RB.

**Figure 5 polymers-12-01854-f005:**
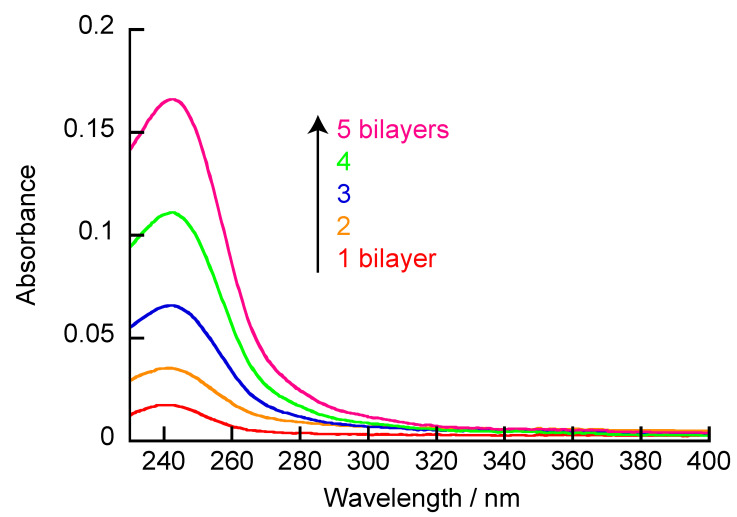
UV-vis absorption spectra for the preparation of (PBA-PAMAM/PVA)_5_ films.

**Figure 6 polymers-12-01854-f006:**
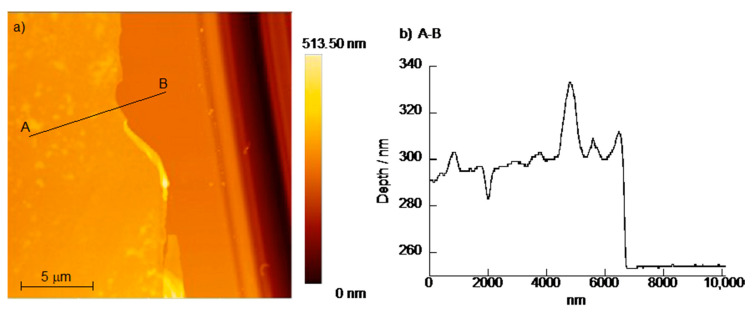
(**a**) atomic force microscopy (AFM) image and (**b**) depth profile of (PBA-PAMAM/PVA)_5_ film.

**Figure 7 polymers-12-01854-f007:**
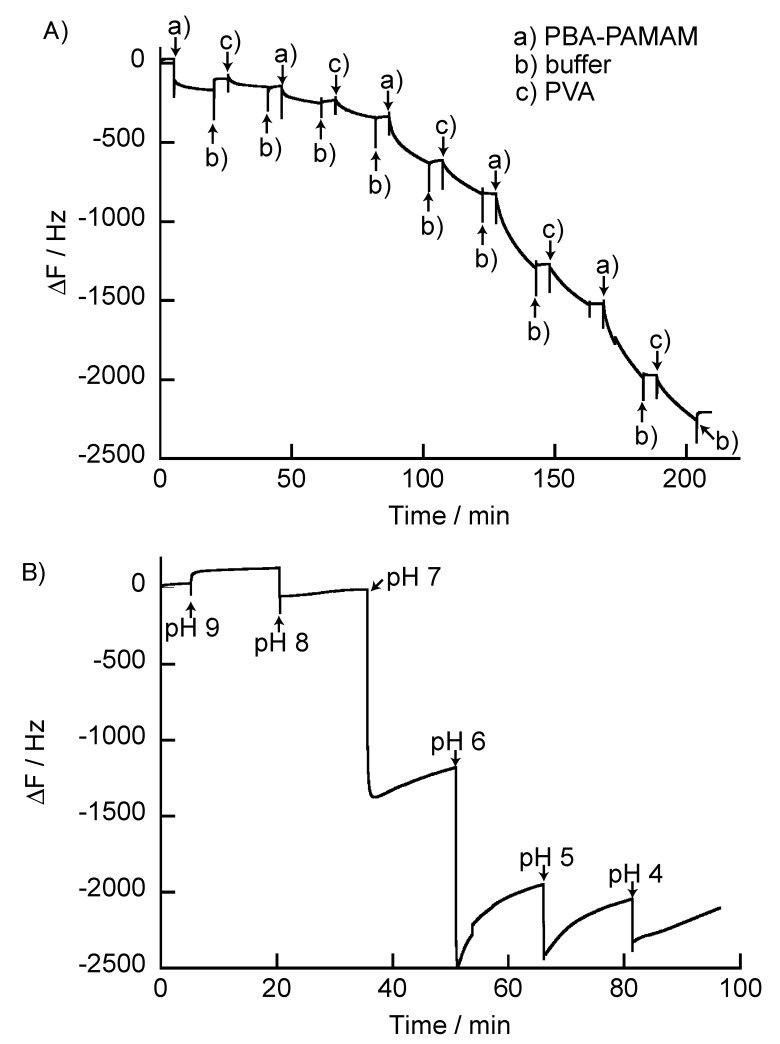
(**A**) frequency change (ΔF) in quartz crystal microbalance (QCM) for the preparation of the (PBA-PAMAM/PVA)_n_ film, and (**B**) frequency changes when the (PBA-PAMAM/PVA)_5_ film is immersed in different pH buffers. The resonator was exposed to (a) 0.1 mg/mL PBA-PAMAM, (b) buffer (pH 9), and (c) 0.1 mg/mL PVA.

**Figure 8 polymers-12-01854-f008:**
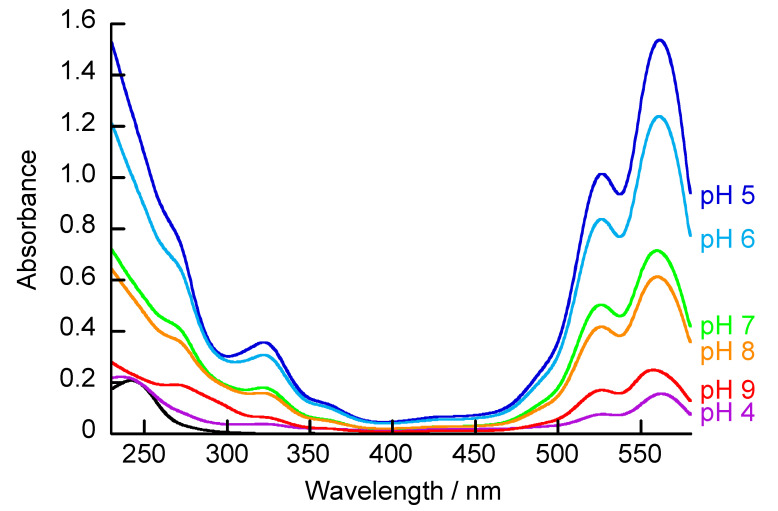
UV-vis absorption spectra of (PBA-PAMAM/PVA)_5_ films after immersion in RB solution at different pHs. The black line is the UV-vis absorption spectrum of the film before immersion in RB solution.

**Figure 9 polymers-12-01854-f009:**
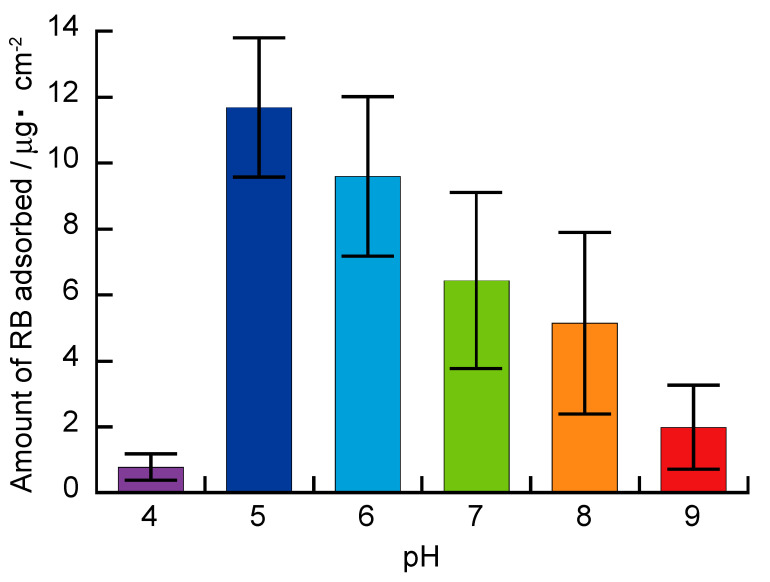
Amount of RB-absorbed on (PBA-PAMAM/PVA)_5_ films after immersion in RB solution with different pHs.

**Figure 10 polymers-12-01854-f010:**
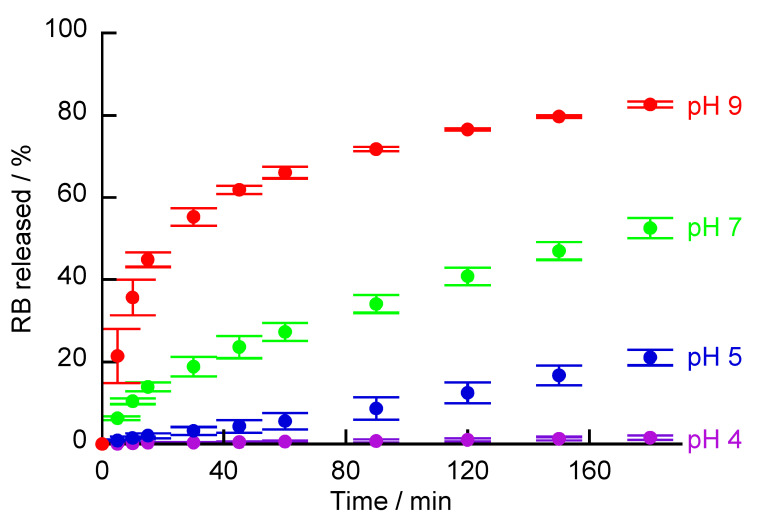
RB released from (PBA-PAMAM/PVA)_5_ films immersed in buffers with various pHs.

**Figure 11 polymers-12-01854-f011:**
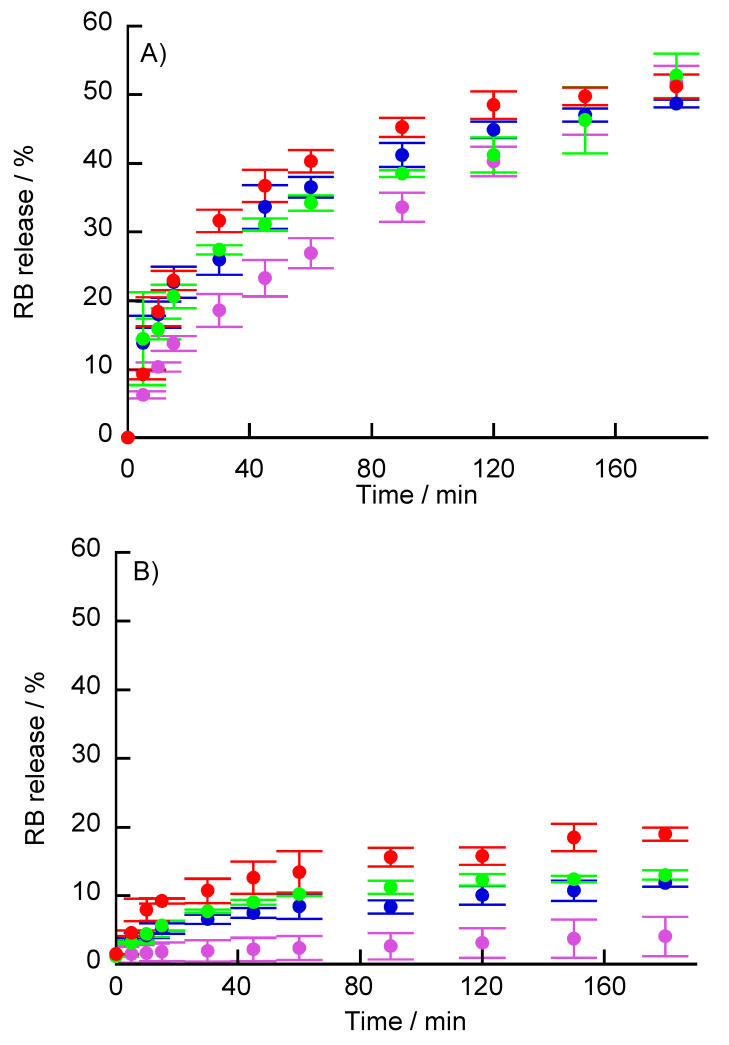
RB released from (PBA-PAMAM/PVA)_5_ films immersed in H_2_O_2_ solutions at (**A**) pH 7 and (**B**) pH 4. The H_2_O_2_ solutions were working buffer solutions of 0 (purple), 1 (blue), 10 (green), and 100 mM (red) H_2_O_2_.

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
