# Peer review of "Adsorption and Release of Rose Bengal on Layer-by-Layer Films of Poly(Vinyl Alcohol) and Poly(Amidoamine) Dendrimers Bearing 4-Carboxyphenylboronic Acid"

_polymers, 2020, doi:10.3390/polym12081854_

Round 1

Reviewer 1 Report

The manuscript describes adsorption and release of rose bengal on layer-by-layer films of poly(vinyl alcohol) and poly(amidoamine) dendrimers bearing 4-carboxyphenylboronic acid.

The paper must be seriously improved before publication.

Major remarks

The increment of absorbance at 250 nm is not in agreement with the numbers of bilayers. Please explain why it happen. After 3 bilayers the absorbance increase non-linear with the number of bilayers. Please explain.

The Lbl bilayers are sensitive to pH. Therefore, the study presented in Figure 8 is incomplete. The absorbance of LbL bilayers at different pH are not the same at different pHs. Therefore, I recommend the study of this dependence and present the spectra in presence of RB corrected.

The results from figures 11 must be clearly presented. These data are statistically significantly different?

The mechanism of RB inclusion / release in/from the LbL film is not clearly described. Please improve.

The practical application is not described. Please add this part.

No data about the selectivity are included in the paper. Please add these information.

Minor remarks

Chemical formula of RB from the Figure 4 is not correct. Please correct it.

H2O2 must be corrected. Please use subscript.

UV-vis is incorrect written. Please correct it.

Author Response

Thank you for your kind suggestion of revision of our manuscript (polymers-859835). We have revised the manuscript according to reviewer’s comments. All revisions made are marked in red in the revised manuscript. Please note that the figure has also been revised. Our responses are as follows.

1) The increment of absorbance at 250 nm is not in agreement with the numbers of bilayers. Please explain why it happen. After 3 bilayers the absorbance increase non-linear with the number of bilayers. Please explain.

[Response] We agree. Added discussion to line 134-138.

2) The Lbl bilayers are sensitive to pH. Therefore, the study presented in Figure 8 is incomplete. The absorbance of LbL bilayers at different pH are not the same at different pHs. Therefore, I recommend the study of this dependence and present the spectra in presence of RB corrected.

[Response] There was not enough explanation. We fixed it as follows. The rinsing and UV-vis spectra measurement solution used the same pH buffer solution as the RB solution (Line 190-191). The lightly adsorbed RB was removed, reducing the effect of the LbL film on pH changes. The pH-sensitive membrane LbL film as a change in the UV-vis spectrum due to the adsorption of RB. These can be considered by the charge of PAMAM and RB in pH change (Line 197-203).

3) The results from figures 11 must be clearly presented. These data are statistically significantly different?
[Response] Added discussion (Line 254-260). There was no difference in the RB release rate due to the difference in H2O2 concentration.

4) The mechanism of RB inclusion / release in/from the LbL film is not clearly described. Please improve.
[Response] We agree. It is considered that the adsorption and release of RB is changed by the protonation of amine in PAMAM. Please check it since I added Line 197-203 and 243-244.

5) The practical application is not described. Please add this part.

[Response]  We agree. Added to 4.Conclusions (Line 289-302.

6) No data about the selectivity are included in the paper. Please add these information.

[Response]  By examining the adsorption and release of RB on the PBA-PAMAM/PVA film, we consider the possibility of a drug reservoir (4.Conclusions). Currently, we are trying a medical drug and it is under investigation.

Minor remarks

1) Chemical formula of RB from the Figure 4 is not correct. Please correct it.

[Response] The structure of RB was corrected.

2)H2O2 must be corrected. Please use subscript.

[Response] We fixed the pointed out part.

3)UV-vis is incorrect written. Please correct it.
[Response] We fixed the pointed out part.

Reviewer 2 Report

This paper explores the use of an LbL method to explore the release of a model Rose Bengal dye within a polymeric system. Generally the paper is well-written, well laid-out and well-presented. With a few minor changes, the paper should be ready for publication. 

Recommended changes:

  • There are some symbols etc., that have not been appropriately subscripted or superscripted. 
  • More details required on the acquisition parameters. For example, UV-vis spectroscopy acquisition parameters. 
  • Fig. SI1: The 'after' line is very hard to see. Perhpas either expand the y-axis or make the line clearer. 

Author Response

Thank you for your kind suggestion of revision of our manuscript (polymers-859835). We have revised the manuscript according to reviewer’s comments. All revisions made are marked in red in the revised manuscript. Please note that the figure has also been revised. Our responses are as follows.

1) There are some symbols etc., that have not been appropriately subscripted or superscripted.

[Response] We agree. We fixed the pointed out part

2) More details required on the acquisition parameters. For example, UV-vis spectroscopy acquisition parameters.

[Response] We agree. We fixed the pointed out part

3) Fig. SI1: The 'after' line is very hard to see. Perhpas either expand the y-axis or make the line clearer.

[Response] Corrected figure.

Reviewer 3 Report

The manuscript „Adsorption and release of rose bengal on layer-by-layer films of poly(vinyl alcohol) and poly(amidoamine) dendrimers bearing 4- carboxyphenylboronic acid” authored by Yoshida et al. reports formation of thin films for adsorption and release of rose Bengal dye. Although the presented research is interesting, the paper suffers from poor English, insufficient characterization of the material associated with wrong/unsatisfactory research design. In my opinion the manuscript is just not ready for publication, thus my recommendation is to reject it, due to the reasons enlisted below:

  1. There are frequent spelling and grammar errors. The text is in some parts unreadable. This obstructs the reading flow and sometimes it is just unclear what the authors meant writing particular sentences. The text should be edited by a native speaker before submission.
  2. The whole text just not describe coherently the story concerning this research. There are not smooth transitions between the discussed results and performed research.
  3. The authors used QCM to prove that they have obtained LBL films. It is unclear form why the kinetics of dye adsorption and release was not studied using this very attractive method. Furthermore, dry thickness measurements by means of AFM or ellipsometry would provide an additional information about percentage mass uptake after dye adsorption.
  4. Measurements of thicknesses should be performed by ellipsometry or AFM  after deposition of single layers to check if the LBL film grows linearly or e.g. exponentially. It is a common approach. Bare QCM measurements are just insufficient.
  5. The presented AFM image seems to be wrongly flattened as LBL layer in shown depth profile is bent unnaturally, which could affect thickness determination.
  6. The authors presented nice QCM measurements presenting behavior of the formed films in the solution with different pH. They speculated that at low pH PVA degrades resulting in swelling of the LBL film. In my opinion, such a fast change in resonant frequency could be rather related to protonation of amine groups in PAMAM and better solvation of the LBL film. The swelling ratio of the layer should be evaluated by thickness determination via AFM or ellipsometry liquid measurements as it is crucial for dye adsorption and release.
  7. I do not understand the choice of the graph format with huge color dots. It is just making the graph unreadable hindering interpretation of the data.
  8. The kinetics of drug (dye) release was not described by any model.
  9. At the end, I have to admit that reading this paper it is unclear in what aspect the developed material is better than the others described in the literature. The authors did not provide any comparison. The obtained results were not somehow referenced to the literature data probably assuming that every reader is an expert in the field or will do literature research by yourself.

Author Response

Thank you for your kind suggestion of revision of our manuscript (polymers-859835). Many of the suggestions are very thought-provoking, and I think they made a great contribution to the revision of the manuscript. We have revised the manuscript according to reviewer’s comments. All revisions made are marked in red in the revised manuscript. Please note that the figure has also been revised. Our responses are as follows.

1) There are frequent spelling and grammar errors. The text is in some parts unreadable. This obstructs the reading flow and sometimes it is just unclear what the authors meant writing particular sentences. The text should be edited by a native speaker before submission.

[Response] We agree. Corrected again. (The address for the english language review is in the acknowledgment.)

2) The whole text just not describe coherently the story concerning this research. There are not smooth transitions between the discussed results and performed research.

[Response] We reported in the following order: “1. Preparation of LbL films, 2. Adsorption of RB on LbL films, 3. Release of RB on LbL films. ” It's hard to understand, so I added a subchapter as well.

3) The authors used QCM to prove that they have obtained LBL films. It is unclear form why the kinetics of dye adsorption and release was not studied using this very attractive method. Furthermore, dry thickness measurements by means of AFM or ellipsometry would provide an additional information about percentage mass uptake after dye adsorption.

[Response] We agree with your suggestion. However, the change in DF derived from RB adsorption was small and it was difficult to consider the change. (The RB dye was colored on the surface of LbL film coated- quartz crystal.) Ellipsometry is suitable for film thickness considerations, but we do not have an ellipsometer. We think that it is possible to consider dye adsorption with AFM. On the other hand, salt precipitation during drying and dye release due to washing could not be ignored. It may be possible if the AFM can be measured in a wet condition. But that wasn't possible with our AFM equipment.

In a future report, we plan to report the adsorption of the drug on the LbL film. At that time, I would like to consider the state change of the LbL films due to drug adsorption by AFM.

4) Measurements of thicknesses should be performed by ellipsometry or AFM  after deposition of single layers to check if the LBL film grows linearly or e.g. exponentially. It is a common approach. Bare QCM measurements are just insufficient.

[Response]  We agree with your suggestion. However, it is difficult to measure the film thickness of one bilayer. (The LbL film growth required for AFM measurements may be insufficient.) On the other hand, LbL film growth can also be considered by UV-vis absorption spectra. (Added to line 134-140.)

5) The presented AFM image seems to be wrongly flattened as LBL layer in shown depth profile is bent unnaturally, which could affect thickness determination.

[Response]  We agree. I re-analyzed the AFM image so that it is flat (Figure 6).

6) The authors presented nice QCM measurements presenting behavior of the formed films in the solution with different pH. They speculated that at low pH PVA degrades resulting in swelling of the LBL film. In my opinion, such a fast change in resonant frequency could be rather related to protonation of amine groups in PAMAM and better solvation of the LBL film. The swelling ratio of the layer should be evaluated by thickness determination via AFM or ellipsometry liquid measurements as it is crucial for dye adsorption and release.

[Response]  We agree with your suggestion. The effect of protonation of amine of PAMAM and solvation with water was added (Line 175-182).

7) I do not understand the choice of the graph format with huge color dots. It is just making the graph unreadable hindering interpretation of the data.

[Response]  We agree. The size of the color dots has been reduced.

8) The kinetics of drug (dye) release was not described by any model.

[Response]  We think it is better to have a model according to your suggestion. However, it is necessary to consider the effects of membrane decomposition (outer inner difference) and diffusion of RB released from the LbL film. Therefore, we think that the preconditions applied to general model formulas (first order etc. ) are different. Rather, it is better to use a model formula such as Korsmeyer-Peppas model, Hixson-Crowell model, or Higuchi model by powdering the thin film. Capsules can be prepared using LbL films. Furthermore, it can be made into powder by drying. We believe that comparisons should be made in this way. We would like to do these as future prospects.

9) At the end, I have to admit that reading this paper it is unclear in what aspect the developed material is better than the others described in the literature. The authors did not provide any comparison. The obtained results were not somehow referenced to the literature data probably assuming that every reader is an expert in the field or will do literature research by yourself.

[Response]  Similar adsorption of RB may be seen when using linear polycations (PAH and PDDA). On the other hand, PAMAM has an advantage in changing the amount of RB adsorbed due to pH.change.

 LbL films using electrostatic interaction have been reported. In the adsorption of drugs on LbL films using films electrostatic interaction, it is necessary to consider the influence between the counter polymer and the drug. We think that the thin film reported in this study is smart because it focuses on the amine protons of PAMAM. In addition, decomposition of thin films requires special polymers (Zwitterionic polymer etc) or hard conditions (0.1M NaOH solution etc). LbL films using diol bonds with PBA have the advantage of being decomposed with H2O2. When combined with oxidases, decomposition of thin films can occur depending on various substrates. There may also be advantages in controlling drug release.

  In this study, we attempted pH and H2O2. responses using RB as a model. Currently, we are trying a medical drug and it is under investigation. We think that it is useful as research in the previous stage.

Round 2

Reviewer 1 Report

The revision is adequate. Therefore, the paper could be published without further modifications.

Author Response

Thank you for your kind suggestions for the revision of our manuscript (polymers-859835).

We have revised the manuscript again.

Reviewer 3 Report

I appreciate the changes and corrections made by the authors. However, I have still some doubts. In my opinion the presented research has a several strong points, but some effects just need more attention and better characterization to improve this paper.

  • The first important issue is not linear growth of LBL films. UV-VIS spectra, as well as QCM measurements show that the film growth is not linear. A number of papers reported exponential film growth (DOI: 10.1039/c0cc03267k; 10.1021/ja7110288). Exponential growth is associated with interdiffusion of building blocks which could result in the creation of some free spaces and pores within the film. It can explain the high capacity of the LBL film presented in this research. That is why I strongly recommend performing of AFM or ellipsometry measurements. It can only improve the paper and help to formulate stronger conclusions. Measurement of topography and thickness of single bilayer is not a challenging task as the AFM resolution in z direction reach single angstroms. See exemplary paper: DOI: 10.1039/c2sm26938d.
  • AFM image in fig. 6 has no thickness scale. It is hardly possible to see any topography details, due to wrong adjustment of the scale.
  • Dye adsorption should be investigated by QCM measurements. The calculated amounts of the adsorbed dye are huge (12 micrograms/cm2 at pH 5), thus I do not understand the argument that the change in frequency is difficult to observe. Assuming the layer density to be in the range of d=1-1.5 g/cm3 and the measured thickness of 5 bilayers h=47 nm, very simple calculations show that the adsorbed mass of LBL film is only 4.7-7 µg/cm2. It means that the layer could adsorb 2 times more mass than initially present on the surface. I think that this should be explained and clarified at least by one complementary method to UV-VIS. SzuwarzyÅ„ski et al. have shown that much smaller adsorption of the dye in thin films could be observed by QCM and AFM methods [DOI:10.1002/chem.201702737].
  • The authors reported that LBL films are substantially decomposed in H2O2, but it was not directly shown. What happened to the layer after long immersion in H2O2? The layer is removed from the surfaces or just the porosity and surface roughness is increased?
  • The authors wrote: “PAMAM protonates tertiary amine” (line 243), „PAMAM protonates primary amines at neutral pH (~7) and protonates primary and tertiary amines at low pH (< 5)”. In what way PAMAM undergoes self-protonation? There are still grammar errors especially in the modified paragraphs.

To sum up, I can support publication of this manuscript when the above arguments will be adequately addressed.  My overall recommendation at this stage is a major revision.  

Author Response

Thank you for your kind suggestions for the revision of our manuscript (polymers-859835). Your suggestions have been helpful and we would like to use your suggestions for future research. Based on your suggestions, we measured QCM and AFM. However, we may not have responded to all suggestions. Our responses are as follows.

1) The first important issue is not linear growth of LBL films. UV-VIS spectra, as well as QCM measurements show that the film growth is not linear. A number of papers reported exponential film growth (DOI: 10.1039/c0cc03267k; 10.1021/ja7110288). Exponential growth is associated with interdiffusion of building blocks which could result in the creation of some free spaces and pores within the film. It can explain the high capacity of the LBL film presented in this research. That is why I strongly recommend performing of AFM or ellipsometry measurements. It can only improve the paper and help to formulate stronger conclusions. Measurement of topography and thickness of single bilayer is not a challenging task as the AFM resolution in z direction reach single angstroms. See exemplary paper: DOI: 10.1039/c2sm26938d.

[Response] We agree with this suggestion. Ellipsometry measurements are considered very important for film thickness and film growth. However, we did not have an ellipsometer. Instead, the surface condition of the LbL film was observed using AFM (Fig. SI1). The surface roughness of the film increased as the number of bilayers increased. This may be related to the non-linear growth of the film. However, dried LbL films may differ from wet LbL films.

2) AFM image in fig. 6 has no thickness scale. It is hardly possible to see any topography details, due to wrong adjustment of the scale.

[Response] We have added a z-axis to Fig. 6a.

3) Dye adsorption should be investigated by QCM measurements. The calculated amounts of the adsorbed dye are huge (12 micrograms/cm2 at pH 5), thus I do not understand the argument that the change in frequency is difficult to observe. Assuming the layer density to be in the range of d=1-1.5 g/cm3 and the measured thickness of 5 bilayers h=47 nm, very simple calculations show that the adsorbed mass of LBL film is only 4.7-7 µg/cm2. It means that the layer could adsorb 2 times more mass than initially present on the surface. I think that this should be explained and clarified at least by one complementary method to UV-VIS. SzuwarzyÅ„ski et al. have shown that much smaller adsorption of the dye in thin films could be observed by QCM and AFM methods [DOI:10.1002/chem.201702737].

[Response] Fig. SI2 shows the change in the DF of the QCM when the quartz resonator coated with the (PBA-PAMAM/PVA)5 film was immersed in RB solutions with various pH. However, when QCM is used, the viscoelasticity changes in a wet multilayer film, and the exact amount of adsorption cannot be discussed. In addition, it is possible that the adsorption of RB on the LbL film changed the solvation of PAMAM, which would result in a complicated change of the viscoelasticity (Lines 216-224).

If QCM is conducted in air, it may be possible to determine the change in the amount of RB adsorption. However, the sample must be rinsed with fresh water to prevent salt precipitation, whereby the amount of RB adsorbed could change; therefore, we avoided measurement with QCM and AFM. However, the release of RB is suppressed at pH 4; therefore, it may be possible to use diluted hydrochloric acid. We would like to consider these techniques for the case where LbL films adsorb drugs.

4) The authors reported that LBL films are substantially decomposed in H2O2, but it was not directly shown. What happened to the layer after long immersion in H2O2? The layer is removed from the surfaces or just the porosity and surface roughness is increased?

[Response] Figure SI4 shows the change in the  DF of the QCM when the (PBA-PAMAM/PVA)5 film and RB-adsorbed (PBA-PAMAM/PVA)5 film are immersed in various H2O2 solutions. A significant increase in DF is observed when immersed in 10 mM H2O2 solution at pH 4 and pH 7 (Fig. SI4A), which indicates decomposition of the LbL films. In addition, the surface of the thin film before and after immersion in the H2O2 solution was observed using AFM (Figs. SI1C and 1D). The surface of the LbL film immersed in H2O2 solution became smooth (the Sa values of the film before and after immersion in H2O2 solution were 9.12 and 1.67 nm, respectively), and most of the film was decomposed (Lines 227-229).

On the other hand, compared with this, the DF change when the RB-adsorbed (PBA-PAMAM/PVA)5 film was immersed in H2O2 solution was low. The adsorption of RB may have suppressed decomposition of the LbL by H2O2. It is possible that RB was deposited on the film due to the reduced solubility of RB under acidic conditions and the adsorption between RB and PAMAM. Therefore, it may be that the dissolution and diffusion of RB was gradually promoted by decomposition of the LbL film in H2O2 (Lines 279-294).

5) The authors wrote: “PAMAM protonates tertiary amine” (line 243), „PAMAM protonates primary amines at neutral pH (~7) and protonates primary and tertiary amines at low pH (< 5)”. In what way PAMAM undergoes self-protonation? There are still grammar errors especially in the modified paragraphs.

[Response] We have added to and corrected the relevant text (Lines 263~268).

Round 3

Reviewer 3 Report

I still have doubts that the amount of the adsorbed dye is overestimated, however, I have to appreciate the additional characterization performed by the authors. I hope that they will include ellipsometry and AFM measurements in the upcoming papers as these should bring better understanding of the LBL film formation and its capacity against drug absorption. My recommendation: accept in the present form.